# Evaluation of Anlotinib Combined with Adriamycin and Ifosfamide as Conversion Therapy for Unresectable Soft Tissue Sarcomas

**DOI:** 10.3390/cancers15030700

**Published:** 2023-01-23

**Authors:** Zuoyao Long, Yajie Lu, Minghui Li, Zhanli Fu, Yunus Akbar, Jing Li, Guojing Chen, Hong-Mei Zhang, Qi Wang, Liangbi Xiang, Zhen Wang

**Affiliations:** 1Xijing Hospital, The Air Force Military Medical University, Xi’an 710000, China; 2General Hospital of Northern Theater Command, Shenyang 110000, China; 3The Third People’s Hospital, Xi’an 710000, China; 4The First Affiliated Hospital, Xinjiang Medical University, Urumqi 830000, China

**Keywords:** unresectable, soft tissue sarcoma, conversion therapy, anti-angiogenesis, combined therapy

## Abstract

**Simple Summary:**

Currently, as a neoadjuvant conversion therapy (NCT), pazopanib combined with neoadjuvant chemoradiotherapy could improve the pathological response of unresectable soft tissue sarcoma (uSTS). However, there is no available evidence of its effectiveness with respect to tumor reduction and surgical margins. In China, anlotinib is a novel tyrosine kinase inhibitor (TKI) that is approved for STS. In this study, adding anlotinib to adriamycin and ifosfamide resulted in an increased rate of tumor regression, surgical conversion and R0 resection in patients with uSTS, particularly for synovial sarcoma and liposarcoma. Severe toxicities are more frequent than monotherapy, but they are controllable.

**Abstract:**

(1) Background: This study investigated the safety and efficiency of adriamycin and ifosfamide combined with anlotinib (AI/AN) as a neoadjuvant conversion therapy in uSTS. (2) Methods: Patients with uSTS were eligible to receive AI/An, including adriamycin (20 mg/m^2^/d) and ifosfamide (3 g/m^2^/d) for the first to the third day combined with anlotinib (12 mg/d) for 2 weeks on/1 week off, all of which combine to comprise one cycle. Surgery was recommended after four cycles of treatment. (3) Results: A total of 28 patients were enrolled from June 2018 to December 2020. The best tumor responses included eight patients with partial responses and 20 with a stable disease. Patients with synovial sarcoma and liposarcoma had a significant decrease in the number of tumors compared with fibrosarcoma (*p* = 0.012; *p* = 0.042). The overall response rate and disease control rate were 28.57% and 100%, respectively. In total, 24 patients received surgery, while the rates of limb salvage and R0 resection were 91.67% (*n* = 22/24) and 87.50% (*n* = 21/24), respectively. Until the last follow-up visit, the mean PFS and RFS were 21.70 and 23.97 months, respectively. During drug administration, 67.87% of patients had grade ≥3 AEs. No treatment-related death occurred. (4) Conclusions: AI/AN followed by surgery showed favorable efficiency and manageable safety in patients with uSTS. A randomized controlled study with a large cohort should be performed for further investigations.

## 1. Introduction

Surgery is the optimum option of patients with local soft tissue sarcoma (STS) with respect to eliminating tumors, which could influence outcomes in terms of recurrence, metastasis and survival [1]. However, for unresectable STS (uSTS), defined as tumors that cannot be resected in order to preserve limb function or intactness, tumor size reduction with chemotherapy or radiotherapy followed by R0 resection—known as neoadjuvant conversion therapy (NCT)—is important to conceive [2]. Considering the diversity of drugs and subtypes, there is still no consensus on the regimens of neoadjuvant chemotherapy [3]. In long-term follow-up visits of randomized clinical trials, the non-inferiority of neoadjuvant histotype-tailored chemotherapy regimens compared to standard regimens in STS was confirmed [4]. To date, despite constant drug inventions, adriamycin and ifosfamide (AI) remain the cornerstones of treatment for STS, as they have been for decades [5]. Noticeably, the current evidence suggested that neoadjuvant chemotherapy could not shrink the tumor’s volume visibly, but it increased tumor necrosis rates and reduced micrometastatic diffusion [6]. Therefore, a novel regimen is urgently needed to improve the NCT of patients with uSTS.

Early research studies on angiogenesis indicated that the growth of tumors was accompanied with increased vascularity [7,8]. Hence, researchers hypothesized that the inhibition of angiogenesis would be an effective strategy for anti-tumor processes [9]. Additionally, anti-angiogenetic tyrosine kinase inhibitors (TKIs) could prune abnormal vessels in tumors and remodel the remaining vessels, resulting in a normalized vasculature [10]. Meanwhile, normalized vessels enhanced the delivery and effectiveness of concurrent therapies, which remind us that combinations with chemotherapy might yield the maximal benefit of TKIs [11]. Based on these theories, tremendous advances have been made with respect to TKIs over the past two decades.

Anlotinib is a novel, orally administrated, multi-targeted TKI that can block the pathway involved in angiogenesis and the proliferation of tumors [12]. As the second-line treatment for advanced STS, anlotinib could improve progression-free survival (PFS) and overall survival (OS) to 5.6 and 12 months, respectively, with manageable toxicity [13]. Preclinical studies demonstrated a potential synergistic interaction between anlotinib and conventional cytotoxic chemotherapy, suggesting that drug combinations could result in the most pronounced tumor inhibition in sarcoma-patient-derived xenograft models [14]. A few clinical studies combining these drugs have been conducted, but none of them focused on the neoadjuvant treatment of STS. Therefore, we present our clinical practice of anlotinib combined with AI (AI/AN) for uSTS. To our knowledge, this is the first study that evaluates the surgical conversion and long-term survival of this novel therapeutic approach.

## 2. Materials and Methods

### 2.1. Study Design and Patients

From 1 June 2018 to 31 December 2020, patients from three hospitals who received AI/AN were included. The inclusion criteria were defined as follows: (1) aged 10–70 years; (2) histologically diagnosed as high-grade STS; (3) protocol defined as chemotherapy-sensitive histologies in the 2013 WHO classification of STS [15]; (4) no previous chemotherapy, radiotherapy or targeted therapy; (5) initially unresectable tumor, including a measurable size of more than 5 cm in the deep fascia, or a size of less than 5 cm but with unclear margins with nerves or vessels; (6) the Eastern Cooperative Oncology Group performance status (ECOG PS) score was 0–2, and the expected survival was more than 3 months.

This study was approved by the Medical Science Committee of the First Affiliated Hospital of Air Force Medical University (Approval Document No.: KY20192016-C-1). Written informed consent was provided by all participants prior to their inclusion in the study.

### 2.2. Procedures

After enrollment, the included patients received oral anlotinib 12 mg once daily at 2 weeks on/1-week off, all of which comprised a cycle; adriamycin (20 mg/m^2^/d) and ifosfamide (3 g/m^2^/d) were administered intravenously for the first to the third day of each cycle. Treatment was suspended when the patient exhibited disease progression, death, unacceptable toxicity, or a refusal of further protocol therapy. Definitive surgery was performed at the end of 4 cycles. Surgical margins were assessed using the R classification system, with R0 being the aim of NCT. In patients without R0 resection, anlotinib maintenance, radiotherapy or other therapies were recommended.

### 2.3. Tumor Response and Outcomes

Because anlotinib can induce internal liquidation and necrosis but retain the diameters of tumor, tumor response was assessed by imaging each of the two cycles according to the RECIST 1.1 criteria [16] and volumetric measurements [17].

According to the RECIST1.1 criteria, a complete response (CR) was defined as the disappearance of the targeted lesion. Partial response (PR) was defined as at least a 30% decrease with respect to the longest diameter compared with the baseline. When a tumor had at least a 20% increase with respect to the longest diameter compared with the smallest result since the beginning of therapy or the development of new lesions, we regarded it as a progressive disease (PD). Stable disease (SD) included tumors that had neither sufficient shrinkage for PR nor a sufficient increase in PD.

Tumor volume was measured using Mimics Medical 20.0 software (Materialise). PR was defined as at least a 64% decrease in volume compared with baseline. When the tumor had an increase in volume of at least 40% compared with the smallest result since the beginning of therapy or the appearance of new lesions, we regarded it as PD. The definitions of CR and SD are similar with the RECIST 1.1 criteria.

The primary endpoints for this study included the rate of surgery and objective response rate (ORR). Secondary endpoints included progression-free survival (PFS) defined as the time from enrolment to PD, relapse-free survival (RFS), the time from R0 resection to progression or recurrence, disease control rate (DCR), limb salvage rate, and R0 resection rate.

### 2.4. Safety

All drug-related adverse events (AEs) were carefully recorded since the beginning of the AI/AN treatments. The severity was assessed according to the CTCAE V5.0. Dose adjustments were carried out according to specifications when grade ≥3 AEs occurred. However, the balance between treatments and AEs should be evaluated by the investigators.

### 2.5. Follow-Up Study

Before administration, baseline medical histories of the patients were collected, including previous treatment, ECOG PS score, vital signs, electrocardiogram and laboratory tests. Imaging examinations (including MRI or CT) within 1 month before drug administration were used as baseline data for the evaluation of target lesions, followed by repeat imaging using the same modality every 6 weeks. Lung CT scans were used to exclude distant metastasis. After surgery, an ultrasonography of the operation area and CT of the lungs were required for follow up.

### 2.6. Statistical Analysis

All statistical analyses were carried out by employing PASW Statistics for Windows (ver. 18.0. Chicago: SPSS Inc., Chicago, IL, USA). Normally distributed data were presented as the mean ± SD. Non-normally distributed data were provided as medians and quartile spacing (M(QR)), with group comparisons made using the Wilcoxon rank-sum test. The PFS and RFS were calculated using the Kaplan–Meier method.

## 3. Results

### 3.1. Patient Characteristics

Between 1 June 2018 and 31 December 2020, 28 patients were enrolled. Patients were followed up for an average of 23.61 ± 7.08 (range: 8–39) months, with no drug-related deaths.

Patient characteristics are summarized in Table 1, which shows that there were 17 males and 11 females with an average age of 39.11 ± 13.46 years. Previous surgery treatments were accepted in 32.14% of patients. The majority of tumors were located in lower limbs (*n*= 17, 60.71%). The most common histological subtypes were fibrosarcoma (*n* = 6, 21.43%), synovial sarcoma (*n* = 6, 21.43%), myxoid liposarcoma (*n* = 6, 21.43%) and undifferentiated pleomorphic sarcoma (*n* = 4, 14.29%). The longest diameter observed in tumors in most patients (*n* = 18, 64.29%) was > 10 cm. (Table 1, Figure 1A).

### 3.2. Tumor Response

Of the 28 patients evaluable for responses at cycle 4, the best tumor response of 8 (28.57%) achieved PR and 20 (71.43%) achieved SD with an ORR of 28.57% (95% CI, 10.7–46.4%). No CR or PD occurred during the treatment; thus, the DCR achieved 100% (Table 2).

In total, 24 patients (85.71%) who received neoadjuvant AI/AN proceeded toward surgery: 22 patients received extended resection and 2 patients required amputation. Of the four patients that had non-operative management, two patients were willing to receive anlotinib maintenance due to the significant tumor regression during NCT, and the other two declined. In total, the R0 resection rate and limb-salvage rate were 87.50% (*n* = 21/24) and 91.67% (*n* = 22/24), respectively.

### 3.3. Tumor Size

Changes in tumor size are shown in Figure 1. Radiographical evaluations showed that the longest diameter with respect to the tumors was reduced in 24 patients (median of 19.36%: from 0.11% to 57.34%) and was increased in 3 patients. According to the measurement results from Mimics 20.0, 25 patients had a tumor volume decrease (median of 28.64% and range from 1.63% to 97.56%) and 2 patients had an increase in volume; in particular, the volume reduction in two patients was beyond 90%. However, the tumor response evaluated with respect to volume was consistent with the RECIST 1.1 criteria, although one patient had a decrease in volume but an opposite change with respect to tumor diameter.

### 3.4. Subgroup Analysis

Subgroup analyses in Figure 2 revealed that patients with synovial sarcoma exhibited significant tumor decreases compared with fibrosarcoma (26.25% vs. 3.10%, *p* = 0.012). A similar advantage was observed in liposarcoma patients (25.32% vs. 3.10%, *p* = 0.042). Tumor responses exhibited no significant differences in age or tumor burden (*p* = 0.203, *p* = 0.413). In particular, in two patients with undifferentiated pleomorphic sarcoma and primitive neuroectodermal tumor, the volume reductions were beyond 90%, and this is unattainable with chemotherapy alone (Figure 3A).

### 3.5. Long-Term Survival

The median follow-up period was 23.5 months (range of 8 to 39 months), and 14 events were observed (Figure 3B, C). The corresponding PFS probabilities at 6, 12 and 24 months were 75.00% (95% CI, 54.60% to 87.22%), 60.71% (95% CI, 40.38% to 75.99%), and 46.56% (95% CI, 26.11% to 64.72%) in all patients, respectively. For patients who received R0 resections, the RFS probabilities at 6, 12 and 24 months were 76.19% (95% CI, 51.93% to 89.33%), 71.43% (95% CI, 47.15% to 86.02%), and 53.03% (95% CI, 28.01% to 72.90%), respectively. In total, the mean PFS and RFS time were 21.70 (95% CI: 16.38, 27.02) and 23.97 (95% CI: 18.00, 29.94) months (Table 2), respectively. At the time of the last follow-up visit, two patients had died due to disease progression, one of which had received R2 resection, anlotinib maintained for 6 months, and the disease progressed after drug withdrawal in a month (the overall survival time of 10 months after surgery). The other patient received R0 resections, with a PFS and overall survival time of 4 and 16 months, respectively.

### 3.6. Safety

During the AI/AN treatment, all patients experienced treatment-related AEs, and most (*n* = 19, 67.86%) had grade ≥3 AEs. No treatment-related death occurred. The NCT-related AEs are listed in Table 3. The most common AEs were leukopenia (22, 78.57%), anorexia (17, 60.71%), fatigue (16, 57.14%), hypertension (12, 42.86%) and oral mucositis (12, 42.86%). NCT-related grade ≥3 AEs included leukopenia (16, 57.14%), hypertension (4, 14.29%) and hand-foot syndrome. All surgeries were performed by the same surgical team, and no surgery-related complications were observed.

Initially, the first patient suffered serious leukopenia after the first cycle, although the tumor reduced from 12.4 cm to 8.3 cm. For safety reasons, we recommended granulocyte colony-stimulating factor (G-CSF) for symptomatic treatment and 25% dose reduction. However, the tumor had no apparent change compared to its previous state, and this was likely due to dose adjustments. Therefore, we suggested that the initial dosage was necessary, and symptomatic treatments should be implemented prior to the management of grade ≥3 AEs if there was no risk to life.

Finally, all patients received an initial dosage except for the first patient, and all AI/AN-related AEs were tolerable or well controlled.

## 4. Discussion

This retrospective study demonstrated that anlotinib combined with adriamycin and ifosfamide could potentially improve surgical conversions in patients with unresected, intermediate-grade or high-grade chemosensitive STS, which is an alternative strategy for these diseases. Furthermore, as far as we know, this is the first study that investigated the efficiency and safety of AI/AN as NCT in uSTS patients with the same regimen.

The growth of tumors is always accompanied with neovascularization. In oncogenesis, tumor cells need vessels via angiogenesis to supply oxygen and nutrients in order to proliferate, with the secretion of several angiogenic factors and chemokines facilitating the stimulation of angiogenesis [18,19]. Based on these theories, bevacizumab, the first angiogenesis inhibitor-targeted vascular endothelial growth factor (VEGF), was approved for cancer therapy in 2004 [11]. Despite initial frustrations and negative clinical trial results, tremendous advances have been made with respect to anti-angiogenetic TKIs over the past two decades [20]. Anlotinib is one of the oral multi-target TKIs with broad-spectrum anti-tumor activities in various solid tumors which could suppress tumor proliferation, migration and metastasis by blocking the VEGFR, platelet-derived growth factor receptor, fibroblast growth factor receptors and c-Kit [21,22]. Based on the potential of anti-STS tumor in a phase I study [23], Chi et al. investigated the efficiency of anlotinib in refractory metastatic STS [13]. A total of 166 patients were included for analysis, with a PFS and overall survival (OS) of 5.6 and 12 months, respectively. Despite the poor DCR of 13%, anlotinib achieved superior PFS compared with other TKIs, including regorafenib and pazopanib. Considering the notably effective and manageable toxicity, anlotinib was recommended as an STS treatment by the Chinese Society of Clinical Oncology in 2019. However, there was insufficient evidence with respect to unequivocally proving that anlotinib monotherapy could shrink tumors to achieve surgical conversion.

TKIs enhanced the effectiveness of concurrent therapies synergistically [24,25,26]. In the experience of Jain R.K et al., TKIs could prune abnormal vessels in tumors and remodel the remaining vessels, resulting in a normalized vasculature [10]. Consequently, normalized vessels could enhance the delivery and effectiveness of concurrent therapies, reminding us that a combination with chemotherapy might yield the maximal benefits of TKIs. The first study focusing on the efficiency of chemotherapy with or without TKI for uSTS was reported by Weiss AR et al. in 2020 [17]. At an interim analysis with 42 patients in this randomized phase 2 trial, a rate of ≥90% with respect to pathological responses was 58% (*n* = 14/24) in the pazopanib group and 22% (*n* = 4/18) in the control group, with a between-group difference of 36.1%. There was no significant difference in tumor responses between the treatment groups, and the efficiency of anlotinib-combined therapies remains controversial. Wang HY et al. reviewed 32 patients with advanced/metastatic STS who received chemotherapy combined with anlotinib in addition to anlotinib maintenance therapy; the PFS was improved to 8.2 months with an ORR and DCR of 34% and 69% [27], respectively. In contrast, a control study in advanced STS revealed no differences in the PFS between patients treated with gemcitabine plus docetaxel and those treated with gemcitabine in addition to anlotinib (*n* = 81, 5.8 months; *n* = 41, 6.8 months; *p* = 0.39), as well as in OS (14.7 vs 13.3 months, *p* = 0.75) [28]. In the current study, anlotinib was selected to combine with AI, the most active regimens, as neoadjuvant therapy for 28 patients with uSTS with the aim of surgical conversion. Four cycles after AI/AN administration, eight patients achieved PR and the others achieved SD. Eventually, the results demonstrated that excellent surgical conversion was achieved in 24 (85.71%) patients. For histological analysis, patients with synovial sarcoma and liposarcoma were more sensitive to AI/AN compared to those with fibrosarcoma.

Another purpose of NCT was to improve adequate resection. However, these results were rarely reported in previous studies. According to the AJCC criteria, a negative margin was defined as exhibiting no microscopic residual disease (R0) and microscopic (R1) or macroscopic (R2) residual diseases at the inked resection margin [29]. The status of surgical margins was an independent adverse factor of prognosis, and it has been reported that patients with positive surgical margins had a 3.3 times greater risk (95% CI, 2.3–4.7) of developing local recurrence compared with negative cases [30]. Moreover, the positive margin was a negative prognostic factor of OS (Hazard ratio = 1.99, 95% CI: 1.15–3.45; *p* = 0.014) [31]. In a retrospective study comprising 181 patients with extreme STS, adequate surgical margins could be beneficial for patients undergoing primary surgery, as well as those who received reoperations with a significantly better post-recurrence survival (*p* < 0.007) and disease-specific survival [29]. Interestingly, there were no differences with respect to disease-specific survival between patients who achieved non-R0 resection after relapse and those with unresectable relapse. Therefore, R0 resection was the primary treatment for STS in order to facilitate a necessary cure [32]. In our research study, 24 (85.71%) patients with uSTS achieved surgical conversion successfully after the administration of AI/AN, with a limb-salvage rate of 91.67%. Three patients underwent non-R0 resection due to an invasion of blood vessels or pelvic viscera. Eventually, patients who received R0 resection had an RFS of 23.97 (95% CI: 18.00, 29.94) months, including nine patients with recurrence or metastasis. PD occurred in two of three patients who received non-R0 resection and three of four patients who refused surgery.

Anlotinib monotherapy exhibited mild toxicities, and this was similar with other TKIs, including a low incidence of triglyceride elevation, hand-foot skin reaction, hypertension, and fatigue [13]. Most of them were grade 1–2 and well tolerated or reversible. In consideration of the superposed toxicities of anlotinib combined with chemotherapy, the AEs in this study were carefully monitored. In our first patient, we observed that a dose reduction could relieve the grade of AEs and lower tumor responses. Thus, we recommended an initial dosage of AI/AN for uSTS with safety guarantees. The most common grade 3 or greater AE was leukocytopenia in 16 (57.14%) patients, which occurred more frequently compared to monotherapy. In order to achieve the expected efficiency, the patients routinely received G-CSF maintenance to prevent and treat leukocytopenia without dose adjustments. In total, 19 patients suffered grade ≥3 AEs, and no death was related to the treatment.

One of the limitations of this study was omitting the pathological response evaluation. Treatment-induced tumor necrosis following neoadjuvant therapy has been proven to be a reliable predictor of outcomes in sarcomas. However, considering the various levels of pathology in three participating centers, we ignored pathological responses in order to avoid biases. Another limitation was the inferior ORR in our study. The ORR of AI/AN in uSTS was only 28.57%, which was lower than 52% in pazopanib combined with chemoradiotherapy [17]. Nevertheless, surgical conversion was successfully achieved in 24 patients and R0 resection in 21 patients. Therefore, we suggested that the success of conversion therapy should be determined by any tumor reduction and clear boundaries. Thirdly, the number of participants in each subtype was limited. As is well known, STS is a group of malignant tumors originating from mesenchymal tissues with high heterogeneity [33]. Thus, the efficiency of AI/AN in different subtypes requires further investigation. Finally, it should be noted that selection biases were unavoidable in this single-arm, retrospective study.

## 5. Conclusions

In conclusion, the addition of anlotinib to AI as neoadjuvant conversion therapy resulted in an increased rate of tumor reduction and surgical conversion, suggesting that this is an alternative combination that can be implemented in patients with unresectable soft tissue sarcoma, particularly for synovial sarcoma and liposarcoma. Severe toxicities are more frequent than monotherapy, but they are controllable. However, a prospective, randomized, controlled trial with a larger cohort of patients should be conducted for further investigation.

## Figures and Tables

**Figure 1 cancers-15-00700-f001:**
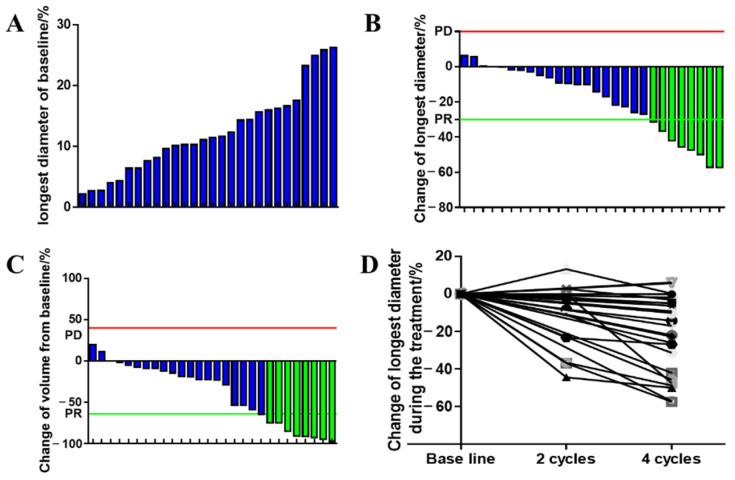
The change in tumor diameters and volume during NCT. (**A**) Longest diameters of tumor at baseline; (**B**) best treatment response changes in the longest tumor diameters; (**C**) best treatment response of tumor volumes; (**D**) changes in the longest tumor diameters at cycle 2 and 4 during AI/AN administration. Blue: tumor response exhibits stable diseases. Green: tumor response exhibits partial responses.

**Figure 2 cancers-15-00700-f002:**
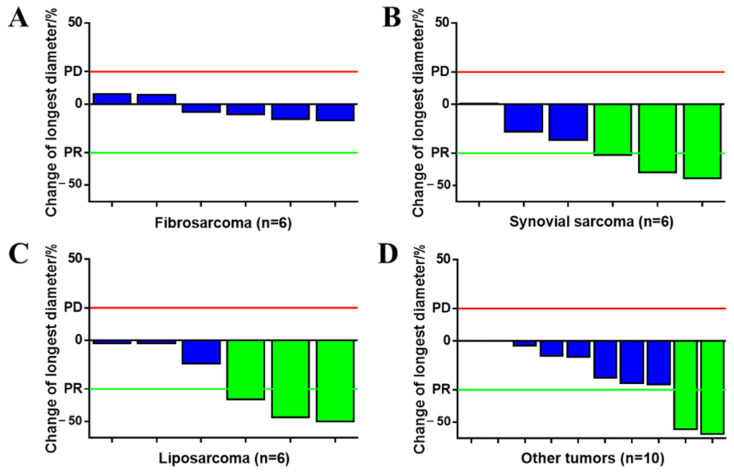
Tumor response of different pathological subtypes. (**A**) Fibrosarcoma (*n* = 6); (**B**) synovial sarcoma (*n* = 6); (**C**) liposarcoma (*n* = 6); (**D**) other sarcomas (*n* = 10). Blue: tumor response exhibits stable disease. Green: tumor response exhibits partial responses.

**Figure 3 cancers-15-00700-f003:**
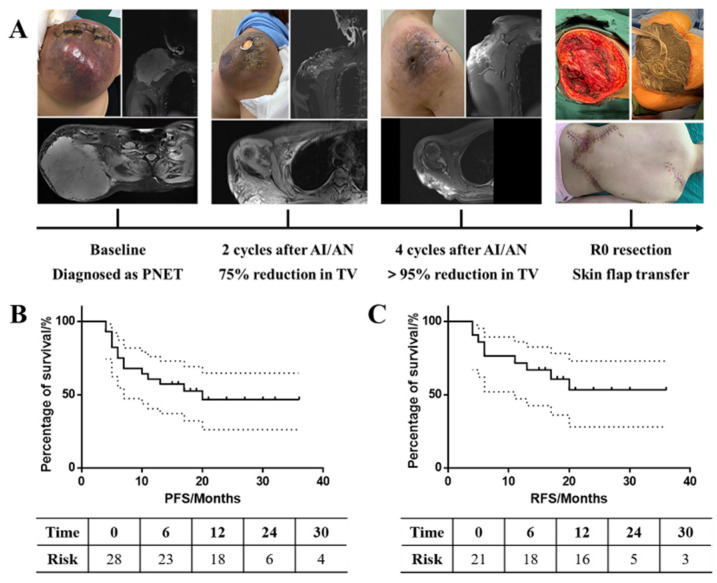
Long-term survival of patients treated with AN/AI. (**A**) Tumor response of the patient with PNET: the reduction in TV was beyond 95%. Up until now, the patient received R0 resections with an RFS of 19 months. (**B**) PFS of all 28 patients; (**C**) RFS of 21 patients who received surgery. PNET, primitive neuroectodermal tumor. TV, tumor volume. AI/AN, adriamycin and ifosfamide combined with anlotinib. PFS: progression-free survival; RFS: relapse-free survival.

**Table 1 cancers-15-00700-t001:** Baseline characteristics of patients.

	*n*	Percentage (%)
**Gender**		
Male	17	60.71%
Female	11	39.29%
**Age (years)**	39.11 ± 13.46 (12–65)	
<30	10	35.71%
31–50	11	39.29%
>50	7	25.00%
**Tumor location**		
Upper limb	7	25.00%
Lower limbs	17	60.71%
Trunk	4	14.29%
**Histology**		
Fibrosarcoma	6	21.43%
Synovial sarcoma	6	21.43%
Liposarcoma	6	21.43%
UPS	4	14.28%
Others	6	21.43%
**Surgery**		
Yes	9	32.14%
No	19	67.86%
**Radiotherapy**		
Yes	1	3.70%
No	27	96.30%
**ECOG PS score**		
0–1	27	96.30%
2	1	3.70%
**Longest** **diameter (cm)**		
<10	10	35.71%
10~20	14	50.00%
>20	4	14.29%

UPS: Undifferentiated pleomorphic sarcoma; ECOG PS score: ECOG performance status score.

**Table 2 cancers-15-00700-t002:** Tumor response, surgical management and long-term survival.

Response	*n*	Percentage (95% CI, %)
CR	0	0
PR	8	28.57 (10.7, 46.4)
SD	20	71.43 (53.6, 89.3)
PD	0	0
**ORR**	28.57 (10.7, 46.4)	
**DCR**	100	
**Surgical margins**	24	
R0	21	87.50
R2	3	12.50
**Surgical type**		
Limb-salvage	22	91.67
Amputation	2	8.33
**Mean PFS/months**	21.70 (16.38, 27.02)	
6 months PFR	75.00 (54.60, 87.22)	
12 months PFR	60.71 (40.38, 75.99)	
**Mean RFS/months**	23.97 (18.00, 29.94)	
6 months RFR	76.19 (51.93, 89.33)	
12 months RFR	71.43 (47.15, 86.02)	

In total, 24 patients received surgical management. Of the four patients receiving non-operative management operations, two patients were willing to receive anlotinib maintenance due to the significant tumor regression in NCT, and the others terminated treatment. CR: Complete response; PR: partial response; SD: stable disease; PD: progressive disease; ORR: overall response rate; DCR, disease control rate; PFS: progression-free survival defined as time from enrollment to progression; RFS (relapse-free survival) was the time from surgery to progression or recurrence.

**Table 3 cancers-15-00700-t003:** Adverse events to AI/AN during the treatment.

	Overall AEs		Grade ≥ 3 AEs	
	*n*	%	*n*	%
Leukocytopenia	22	78.57	16	57.14
Anorexia	17	60.71	0	0
Fatigue	16	57.14	0	0
Hypertension	12	42.86	4	14.29
Oral mucositis	12	42.86	0	0
Hand-foot syndrome	11	39.29	2	7.14
Anemia	8	28.57	0	0
Abnormal liver function	7	25.00	0	0
Nausea and vomiting	7	25.00	1	3.57
Weight loss	5	17.86	0	0
Thrombocytopenia	5	17.86	0	0
Decrease in total protein	5	17.86	0	0
Diarrhea	5	17.86	0	0
Gingival bleeding	5	17.86	1	3.57
Hypothyroidism	4	14.29	0	0
Proteinuria	4	14.29	1	3.57
Skin pigmentation	2	7.14	0	0
Alopecia	2	7.14	0	0
Hematuresis	2	7.14	0	0
Delayed wound healing	2	7.14	0	0
Throat pain	1	3.57	0	0

All patients experienced treatment-related AEs, and most (*n* = 19, 67.86%) had grade ≥ 3 AEs. We suggested that an initial dosage was necessary, and symptomatic treatments were carried out previously to manage grade ≥3 AEs if there were no risks to life. No treatment-related death occurred. AI/AN, adriamycin and ifosfamide combined with anlotinib.

## Data Availability

The data presented in this study are available on request from the corresponding author.

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
