# Peer review of "Evaluation of Anlotinib Combined with Adriamycin and Ifosfamide as Conversion Therapy for Unresectable Soft Tissue Sarcomas"

_cancers, 2023, doi:10.3390/cancers15030700_

Round 1
Reviewer 1 Report
The submitted paper" Evaluation of anlotinib combined with adriamycin and ifosfamide as conversion therapy for unresectable soft tissue sarcomas " is an appreciable attempt to add at the usual combination DOXO + IFO in neoadjuvant (NA) setting the antiangiogenic drug Anlotinib, but some weaknesses of the text should be revisited
1) Abstract includes some phrases from the Authors' instructions that must be deleted.
2) The introduction is confused mainly in the first part: NA threrapy is not only prescribed to decrease the volume of the tumor, but either to reduce the micrometastatic diffusion, or to assess the chemosensitivity of one specific subtype of STS. The final end point should be the improvement of the survival.
None of these aspects are considered in the introduction.
Other points to be analyzed:
Anlotinib is synergistic with chemotherapy or neutral?
Why was Anlotinib administered as maintenance in a Phase II study without control arm?
Are toxicities increased by the combination of Anlotinib with DOXO + IFO?
3) Patients and methods: it is strange to include Patients from 10 ( ??) to 70 years old. As far as I know all the studies in adult form of STS include people from 18 to 70 or 75 years Childrens' STS are usually different in histology from adults' ones.
Which are the" chemosensitive STS" (WHO 2013 classification) as stated in the text? Why PNET tumors were included as reported in figure 3? In general such histology is not considered a tipical adult STS form and need a specific study.
Furthermore, fibrosarcoma ( 6 cases) is not very sensitive to chemotherapy.
The discussion is thorough and detailed, but confused and include too many aspects not presented in the Introduction.
a)The activity of anlotinib in STS, has contrasting results.
How can the Authors state that "anlotinib combined with adriamycin and ifosfamide could improve surgical conversion in patients with unresected, intermediate grade or high-grade chemosensitive STS" without a confrontation arm?
Which are the basis of the statement that this combination "is a novel strategy in these diseases" without a randomized comparative arm?
b) As rightly stated by the Authors , this study doesn't evaluate the pathological response of the resected samples.
Pathological response should be always considered in NA treament in every type of tumors, mostly in STS.
Author Response
Dear editors and Reviewer 1,
We thank you very much for giving us an opportunity to revise our manuscript, we appreciate editors and reviewers very much for their positive and constructive comments and suggestions on our manuscript entitled “Evaluation of anlotinib combined with adriamycin and ifosfamide as conversion therapy for unresectable soft tissue sarcomas” (cancers-2122852).
We have studied reviewer’s comments carefully and have made revision which marked using the underline feature in the revised version. We have tried our best to revise our manuscript according to the comments.
We would like to express our great appreciation to you and reviewers for comments on our paper. Looking forward to hearing from you.
Thank you and best regards.
Yours sincerely,
Corresponding author:
Zhen Wang, MD, Department of Orthopaedic Surgery, Xijing Hospital, the Air Force Medical University
Address: No. 127 Changle West Road. Xi’an, Shanxi, China
Email: [email protected]; Tel: +86 029-84775281
Response Table to Reviewers (cancers-2122852)
Evaluation of anlotinib combined with adriamycin and ifosfamide as conversion therapy for unresectable soft tissue sarcomas
Numbered Comments |
Author Response |
Reviewers Comments: |
|
1) Abstract includes some phrases from the Authors' instructions that must be deleted. |
Thanks for your advice. Sorry for the mistake when we used the templet from Authors’ instructions. We have deleted the spare phrases as advised (Page 1, line 36).
|
2) The introduction is confused mainly in the first part: NA threrapy is not only prescribed to decrease the volume of the tumor, but either to reduce the micrometastatic diffusion, or to assess the chemosensitivity of one specific subtype of STS. The final end point should be the improvement of the survival. None of these aspects are considered in the introduction. |
We are grateful for the suggestion. As you have mentioned, NA therapy could not only decrease the volume of tumor, but also reduce the micrometastatic diffusion. However, a meta-analysis indicated that the overall relative risk was 0.396 for local recurrence in STS excised with complete margins as compared to incomplete margins (Milovancev et.al, PMID: 30953384). Therefore, for patients with unresectable STS, we considered that the priority was transferring “unresectable” to “resectable”. This study was an explorative treatment of NA therapy with AI/AN for unresectable STS, which was aimed to preliminarily evaluate the efficiency in tumor reduction and surgical conversion. On the other hand, a long-term follow-up of randomized clinical trials revealed the non-inferiority of neoadjuvant histotype-tailored chemotherapy regimen compared to standard regimen in STS. Hence, we recommended adriamycin and ifosfamide as chemotherapy regimen (Page 2, Line 54-57). Based on the satisfying results of tumor reduction and surgical conversion in this study, a prospective controlled trial with larger cohort was recommend to further investigation and evaluation of long-term survival. According to your suggestion, we added the part of “reduce the micrometastatic diffusion” (Page 2, Line 60-61). |
Anlotinib is synergistic with chemotherapy or neutral? Why was Anlotinib administered as maintenance in a Phase II study without control arm? Are toxicities increased by the combination of Anlotinib with DOXO + IFO? |
Thank you for the questions. 1. Anlotinib is synergistic with chemotherapy. In experience of Jain R.K et al, TKIs could prune the abnormal vessels in tumor and remodel the remaining vessels, resulting in a normalized vasculature. Consequently, normalized vessels could enhance the delivery and effectiveness of concur-rent therapies, reminding us that combination with chemotherapy might yield maxi-mal benefit of TKIs (Jain R.K et al, PMID: 23669226. Page 9, Line 269-273). 2. Between May 2013 and May 2015, Yihebali Chi had performed the Phase II study of anlotinib in patients with refractory metastatic STS, which included 166 patients. In the inclusion criteria, eligible patients were required to progress after anthracycline-based first-line chemotherapy and be naïve from antiangiogenic agents. Therefore, there had no need of control group to compare the difference between anlotinib and traditional chemotherapy (Yihebali Chi, PMID: 29895706). 3. Obviously, there had superposed toxicities of anlotinib combined with chemotherapy. In 2020, Weiss AR et al reported a randomized phase 2 trial, the most common grade 3-4 AEs were leukopenia (43%), neutropenia (41%) and febrile neutropenia (9%) in pazopanib group, and neutropenia (41%) and febrile neutropenia (9%) in control group (Weiss AR et al, PMID: 32702309). More patients had serious AEs in pazopanib group. Similarly, 67.87% patients had grade≥3 AEs in our study, which was more frequent than the result of anlotinib phase 2 trial (Yihebali Chi, PMID: 29895706, Page 314-323). |
3) Patients and methods: it is strange to include Patients from 10 ( ??) to 70 years old. As far as I know all the studies in adult form of STS include people from 18 to 70 or 75 years. Childrens' STS are usually different in histology from adults' ones. |
Thanks for your kindly advise. We have carefully reviewed the relevant literatures, and most of these studies were focused on adults (older than 18 years old). However, the ARST1321 study had enrolled eligible adults (age≥18 years) and children (age between 2 and <18 years), for unresectable STS receiving preoperative chemoradiotherapy with or without pazopanib. Especially, there had no difference of regimens between adults and children, except the dose of pazopanib. Finally, paediatric and adult patients had a similar number of grade 3 and 4 toxicity, and no difference in pathological response (Weiss AR et al, PMID: 32702309). In consideration of the results of ARST1321, we enrolled the patients aged from 10 to 70. And as reported by previous studies, STS commonly occurred in adults, and only one patient was under 18 years old (aged 13). |
Which are the" chemosensitive STS" (WHO 2013 classification) as stated in the text? Why PNET tumors were included as reported in figure 3? In general such histology is not considered a tipical adult STS form and need a specific study. Furthermore, fibrosarcoma ( 6 cases) is not very sensitive to chemotherapy.
|
Thanks for your reminder. 1. Chemosensitive STS (WHO 2013 classification) was reported by Jo VY et al in 2013, including adipocytic tumors, fibroblastic / myofibroblastic tumors, so-called fibrohistiocytic tumors, smooth muscle tumors, pericytic tumors, skeletal muscle tumors, vascular tumors and chondro-osseous tumors. We have added the correct reference in manuscript (Jo VY et al, PMID: 24378391, Page 2, Line 89). 2. It’s no doubt that PNET is commonly occurred in children and adolescent. Only one PNET patient was enrolled in our study. However, the tumor reduction (beyond 95%) in this adult patient (aged 30 years) was rare in clinical practice. Even though the result may not be universal, it was worth trying TKI combined with chemotherapy for patients with refractory PNET, either adult or children. 3. Fibrosarcoma had moderate sensitivity to chemotherapy, but good response to anlotinib. In phase 2 trial of anlotinib, 18 patients with advanced fibrosarcoma had a PFR12weeks, median PFS and OS of 81%, 5.6 and 12 months, which was comparable to average results (68%, 5.6 and 12 months). And in ARST1321, good pathologic response was observed in fibrosarcoma. Therefore, we hypothesized that the synergistic effect of anlotinib and chemotherapy could have good response to fibrosarcoma. All tumors in this study were unresectable because of unclear margin or closing to major vessels, and most of them were beyond 100 cm3. Finally, fibrosarcoma had clear margins with surrounding tissue after AI/AN treatment, although the tumor reduction was not obvious. Finally, 3 patients received R0 resection (1 amputation), 1 of R2 resection, 2 patients refused surgery. |
The discussion is thorough and detailed, but confused and include too many aspects not presented in the Introduction. a) The activity of anlotinib in STS, has contrasting results. How can the Authors state that "anlotinib combined with adriamycin and ifosfamide could improve surgical conversion in patients with unresected, intermediate grade or high-grade chemosensitive STS" without a confrontation arm? Which is the basis of the statement that this combination "is a novel strategy in these diseases" without a randomized comparative arm? |
We are grateful for the suggestion. Indeed, our expression was too strong and certain in the part of discussion, which seemed that the findings of the study couldn’t support our conclusion. In phase I study, anlotinib showed promising antitumor potential against many types of tumor such as colon adenocarcinoma, non-small cell lung cancer, renal clear cell cancer, medullary thyroid carcinoma and STS (PMID: 27716285). Yihebali Chi et al had performed phase II study to thoroughly investigate the efficiency of anlotinib in refractory STS, including UPS, LPS, LMS, FS, ASPA and CCS. A total of 166 patients were included in the final analysis with the PFR12 weeks, ORR, median PFS and median OS of 68%, 13%, 5.6 and 12 months, respectively (Yihebali Chi, PMID: 29895706). Based on these promising results, we designed this study to preliminarily investigate the synergistic effect of AI/AN on tumor reduction and limb salvage surgery for patients with unresectable STS. In our experience and related reports, traditional chemotherapy had limited efficiency in tumor reduction, although could reduce the micrometastatic diffusion. However, the aim of this study was to explore the potential of tumor reduction to achieve surgical conversion and R0 resection, which is the cornerstone to cure STS. According to our plan, we are designing a multicentre, randomized trial to further investigate the efficiency and safety of AI/AN in patients with unresectable STS, including pathological response evaluation and long-term survival. We have modified this part according to your suggestion: “This retrospective study demonstrated that anlotinib combined with adriamycin and ifosfamide could potentially improve surgical conversion in patients with unresected, intermediate-grade or high-grade chemosensitive STS, which is an alternative strategy in these diseases.” (Page 8, Line 245-248) |
b) As rightly stated by the Authors, this study doesn't evaluate the pathological response of the resected samples. Pathological response should be always considered in NA treatment in every type of tumors, mostly in STS. |
Thank you for underlining this deficiency. Inevitably, one of the most important limitations in our study was lacking of pathological response evaluation. Different from data analysis, it’s difficult to evaluate pathological response with same standard in these three centers. In this study, patients were treated using the same protocol. All image data including tumor volume, tumor diameters and following-up, were collected and analyzed by the same group. We considered that there had no significant bias in data analysis. However, pathological response evaluation in our study was performed separately in three hospitals. The results were widely restricted to the levels of pathology, time from tumor resection to pathological evaluation and sites of sampling. Considering the uncontrollable bias, we didn’t describe these results in the manuscript. Totally, 10, 9 and 9 patients were treated in three hospitals respectively. The number of patients with a 90% pathological response or higher was 5 patients (55.56%, n=5/9, one didn’t receive surgery) in hospital 1, 3 (37.50%, n=3/8, one didn’t receive surgery) in hospital 2 and 5 (71.43%, n=5/7, two didn’t receive surgery) in hospital 3. Good response was observed in 13 (54.17%) of 24 patients, which was similar to the results of ARST1321 study. But in order to ensure the accuracy and objectivity of this study, we ignored the pathological response evaluation. We are preparing a randomized clinical trial to further investigate the pathological response and long-term of AI/AN in patients with unresectable STS. |

Reviewer 2 Report
Overall, this is an interesting approach and while I agree that the combination of TKIs with cytotoxic chemotherapy is likely to produce improved necrosis rates in a synergistic manner, the study was not designed to show an improvement. Ultimately, this study shows that in a retrospective analysis of a select group of patients, the combination was tolerated and seemed to produce a response. This response cannot be attributed to the anlotinib over the adriamycin/ifosfamide. This question requires a randomized control trial with a head-to-head comparison in order to demonstrate a benefit over conventional therapy. In addition, it does not appear that radiation therapy was utilized which is standard of care for many of these sarcomas (based on size, grade, and histology) and it is the timing of radiation therapy with many TKIs that can be problematic. Overall, I think the conclusions over exaggerate the findings of the study.
Author Response
Dear editors and Reviewer 2,
We thank you very much for the time you spent reviewing our manuscript. And we appreciate editors and reviewers very much for your comments and suggestions on our manuscript entitled “Evaluation of anlotinib combined with adriamycin and ifosfamide as conversion therapy for unresectable soft tissue sarcomas” (cancers-2122852). Undoubtedly, we agree with you on the deficiency of our manuscript, which reducing the credibility of the results.
Comments and Suggestions for Authors
Overall, this is an interesting approach and while I agree that the combination of TKIs with cytotoxic chemotherapy is likely to produce improved necrosis rates in a synergistic manner, the study was not designed to show an improvement. Ultimately, this study shows that in a retrospective analysis of a select group of patients, the combination was tolerated and seemed to produce a response. This response cannot be attributed to the anlotinib over the adriamycin/ifosfamide. This question requires a randomized control trial with a head-to-head comparison in order to demonstrate a benefit over conventional therapy. In addition, it does not appear that radiation therapy was utilized which is standard of care for many of these sarcomas (based on size, grade, and histology) and it is the timing of radiation therapy with many TKIs that can be problematic. Overall, I think the conclusions over exaggerate the findings of the study.
However, we considered that there still have some highlights in our study which was worthy of further investigation. As previous study reported, patients with positive surgical margins had 3.3 times greater risk (95%CI, 2.3-4.7) of developing local recurrence compared with negative cases, which indicated that the status of surgical margins was an independent adverse factor of prognosis. Therefore, how to achieve R0 resection on patients with unresectable STS was an important process to facilitate a necessary cure. But in our experience and related reports, traditional chemotherapy had limited efficiency in tumor reduction, although could reduce the micrometastatic diffusion. Therefore, the aim of this study was to explore the potential of tumor reduction to achieve surgical conversion and R0 resection, which is the basis to perform a randomized trial.
In phase I study, anlotinib showed promising antitumor potential against many types of tumor such as colon adenocarcinoma, non-small cell lung cancer, renal clear cell cancer, medullary thyroid carcinoma and STS (PMID: 27716285). Yihebali Chi et al had performed phase II study to thoroughly investigate the efficiency of anlotinib in refractory STS, including UPS, LPS, LMS, FS, ASPA and CCS. A total of 166 patients were included in the final analysis with the PFR12 weeks, ORR, median PFS and median OS of 68%, 13%, 5.6 and 12 months, respectively (PMID: 29895706). Based on these promising results, we designed this study to preliminarily investigate the synergistic effect of AI/AN on tumor reduction and limb salvage surgery for patients with unresectable STS. According to our plan, we are designing a multicentre, randomized trial to further investigate the efficiency and safety of AI/AN in patients with unresectable STS, including pathological response evaluation and long-term survival.
Inevitably, one of the most important limitations in our study was lacking of pathological response evaluation. Different from data analysis, it’s difficult to evaluate pathological response with same standard in these three centers. The results were widely restricted to the levels of pathology, time from tumor resection to pathological evaluation and sites of sampling. In our study, the number of patients with a 90% pathological response or higher was 5 patients (55.56%, n=5/9, one didn’t receive surgery) in hospital 1, 3 (37.50%, n=3/8, one didn’t receive surgery) in hospital 2 and 5 (71.43%, n=5/7, two didn’t receive surgery) in hospital 3. Good response was observed in 13 (54.17%) of 24 patients, which was similar to the results of ARST1321 study (PMID: 32702309). But in order to ensure the accuracy and objectivity of this study, we ignored the pathological response evaluation.
Indeed, our expression was too strong and certain in the part of discussion, which seemed that the findings of the study couldn’t support our conclusion. We tried our best to modified our manuscript which would not influence the content and framework of the paper. We appreciate for editors and reviewers warm work earnestly, and hope the answer and correction will meet a new opportunity.
Once again, we would like to express our great appreciation to you for comments on our paper. Looking forward to hearing from you.
Thank you and best regards.
Yours sincerely,
Corresponding author:
Zhen Wang, MD, Department of Orthopaedic Surgery, Xijing Hospital, the Air Force Medical University
Address: No. 127 Changle West Road. Xi’an, Shanxi, China
Email: [email protected]; Tel: +86 029-84775281

Round 2
Reviewer 1 Report
The present version of the paper is well presented and the design of the study now is clear. The conclusion even debatable are acceptable.